# Estimation of Water Quality Parameters through a Combination of Deep Learning and Remote Sensing Techniques in a Lake in Southern Chile

**Lien Rodríguez-López** [1,*], **David Bustos Usta** [2], **Iongel Duran-Llacer** [3], **Lisandra Bravo Alvarez** [4], **Santiago Yépez** [5], **Luc Bourrel** [6], **Frederic Frappart** [7] and **Roberto Urrutia** [8]

[1] Facultad de Ingeniería, Arquitectura y Diseño, Universidad San Sebastián, Lientur 1457, Concepción 4030000, Chile

[2] Facultad de Oceanografía, Universidad de Concepción, Concepción 4030000, Chile; davidbustos@udec.cl

[3] Hémera Centro de Observación de la Tierra, Facultad de Ciencias, Ingeniería y Tecnología, Universidad Mayor, Camino La Pirámide 5750, Huechuraba 8580745, Chile; iongel.duran@umayor.cl

[4] Department of Electrical Engineering, Universidad de Concepción, Edmundo Larenas 219, Concepción 4030000, Chile; lisanbravo@udec.cl

[5] Department of Forest Management and Environment, Faculty of Forestry, Universidad de Concepcion, Calle Victoria, Concepción 4030000, Chile; syepez@udec.cl

[6] Géosciences Environnement Toulouse, UMR 5563, Université de Toulouse, CNRS-IRD-OMP-CNES, 31000 Toulouse, France; luc.bourrel@ird.fr

[7] INRAE, Bordeaux Sciences Agro, UMR 1391 ISPA, Université de Bordeaux, 33604 Talence, France; frederic.frappart@inrae.fr

[8] Facultad de Ciencias Ambientales, Universidad de Concepción, Concepción 4030000, Chile; rurrutia@udec.cl

* Correspondence: lien.rodriguez@uss.cl; Tel.: +56-999168115

**Abstract:** In this study, we combined machine learning and remote sensing techniques to estimate the value of chlorophyll-a concentration in a freshwater ecosystem in the South American continent (lake in Southern Chile). In a previous study, nine artificial intelligence (AI) algorithms were tested to predict water quality data from measurements during monitoring campaigns. In this study, in addition to field data (Case A), meteorological variables (Case B) and satellite data (Case C) were used to predict chlorophyll-a in Lake Llanquihue. The models used were SARIMAX, LSTM, and RNN, all of which showed generally good statistics for the prediction of the chlorophyll-a variable. Model validation metrics showed that all three models effectively predicted chlorophyll as an indicator of the presence of algae in water bodies. Coefficient of determination values ranging from 0.64 to 0.93 were obtained, with the LSTM model showing the best statistics in any of the cases tested. The LSTM model generally performed well across most stations, with lower values for MSE ($<0.260$ $(\mu g/L)^2$), RMSE ($<0.510$ ug/L), MaxError ($<0.730$ $\mu g/L$), and MAE ($<0.442$ $\mu g/L$). This model, which combines machine learning and remote sensing techniques, is applicable to other Chilean and world lakes that have similar characteristics. In addition, it is a starting point for decision-makers in the protection and conservation of water resource quality.

**Keywords:** water quality; chlorophyll; remote sensing; deep learning; Chile; lakes

## 1. Introduction

Eutrophication is a phenomenon that occurs in lakes when excess nutrients such as phosphorus and nitrogen accumulate in the water [1,2]. This leads to the growth of algae and other aquatic plants, which can deplete oxygen levels in water and create harmful algal blooms [3,4]. The effects of eutrophication can be devastating to lake ecosystems, leading to the death of fish and other aquatic species [5,6]. Interventions to reduce eutrophication in lake watersheds include decreasing fertilizer use in nearby agriculture, limiting the discharge of raw sewage into the lake, and introducing species that consume excess nutrients, such as carp and tilapia [7,8]. In addition, other practices carried out by lake managers such

as dredging and aeration are causing an increase in oxygen levels and reducing nutrient concentrations [9,10].

Chlorophyll-a (Chl-a) is a green pigment found in the chloroplasts of algae, plants, and some bacteria that is responsible for capturing light energy during photosynthesis [11,12]. It is often used as an indicator of algae presence because it is a primary photosynthetic pigment found in all types of algae [13,14]. The concentration of Chl-a is directly proportional to the number of algae present in a water sample, making it a useful tool for monitoring algal growth and detecting harmful algal blooms [15,16]. Additionally, Chl-a can be used to estimate the primary productivity of aquatic ecosystems, which is an important factor for understanding the health and functioning of these systems [17,18].

Remote sensing techniques, combined with artificial intelligence models, have revolutionized the way scientists study, and manage the Earth's natural resources [19,20]. These techniques involve the use of sensors to collect data remotely, often from satellites, aircraft or drones [21–23]. By using artificial intelligence algorithms to analyze the collected data, researchers can gain insights into environmental patterns and make predictions about future trends [24,25]. With the help of machine learning models, scientists can develop early warning systems to detect harmful algal blooms, helping to mitigate the negative effects of eutrophication on lake ecosystems [26–28]. Moreover, multiple remote sensing studies have been used to monitor water quality in lakes and identify changes in nutrient concentrations that could lead to eutrophication [29–32]. The combination of remote sensing and artificial intelligence provides a powerful tool for understanding and managing complex environmental systems.

Chile has several lake districts, from north to south: the district of Altiplanic Lakes, Nabuelbutan Lakes, Araucanian Lakes, Chiloe Lakes, and Nordpatagonian Lakes or Paine Towers. Araucanian lakes stand out for their economic, social, and environmental importance. Lake Llanquihue is the largest lake in this chain and there is little scientific knowledge about aspects of its water quality, which is why it was selected for study in a previous investigation and specifically in the present one. The objective of this study is to contribute through combined techniques of remote sensing and machine learning to develop early tools for monitoring lakes in the follow-up of algal bloom phenomena. For this, we will follow the following specific objectives: (i) analyze the behavior of the physicochemical and biological variables best related to algal bloom events during the period 1989–2021; (ii) train artificial intelligence models with real in situ data of limnological and meteorological variables and data from Landsat satellite image sources and (iii) estimate the concentration of chlorophyll-a in the lake for seasons of the year where monitoring data are not available and validate these results with data from monitoring campaigns.

## 2. Materials and Methods

### 2.1. Site Description

Lake Llanquihue (41°08′S and 72°47′W) is a large freshwater lake located in southern Chile, in the Los Lagos Region [33]. The lake is located at an altitude of 51 m above sea level (m.a.s.l.) and has an area of approximately 870 km$^2$ (Figure 1), making it one of the largest lakes in Chile [34]. It is also one of the most emblematic natural landmarks in the region and is surrounded by the impressive backdrop of the Andes Mountains. The lake is fed by several rivers and streams, such as the Maullín and Petrohué rivers, which flow into the lake from east and west, respectively. The mouth of Lake Llanquihue is the Maullín River that flows into the Pacific Ocean. The lake has a maximum depth of 317 m and an average depth of 207 m, making it one of the deepest lakes in South America [35]. Lake Llanquihue is surrounded by several active volcanoes, such as Osorno, Calbuco and Puntiagudo, which contribute to the region's rugged and dramatic landscape [36]. The lakeshore is characterized by beaches, cliffs, and rocky outcrops, with a variety of flora and fauna present in the surrounding forests and wetlands. The climate around Lake Llanquihue is classified as temperate oceanic, with mild temperatures and high humidity throughout the year. The average annual temperature is around 11 °C, with averages ranging from

5 °C in winter to 16 °C in summer [37]. The region is also known for its frequent rainfall, especially during the winter months, which contributes to the lush vegetation and fertile soil of the surrounding area. Overall, Lake Llanquihue is an impressive natural feature and the product of the dynamic geological and climatic forces that have shaped the landscape of southern Chile. Its deep, crystal-clear waters, surrounded by volcanoes and green forests, make it a popular destination for outdoor activities such as hiking, fishing, and kayaking [38].

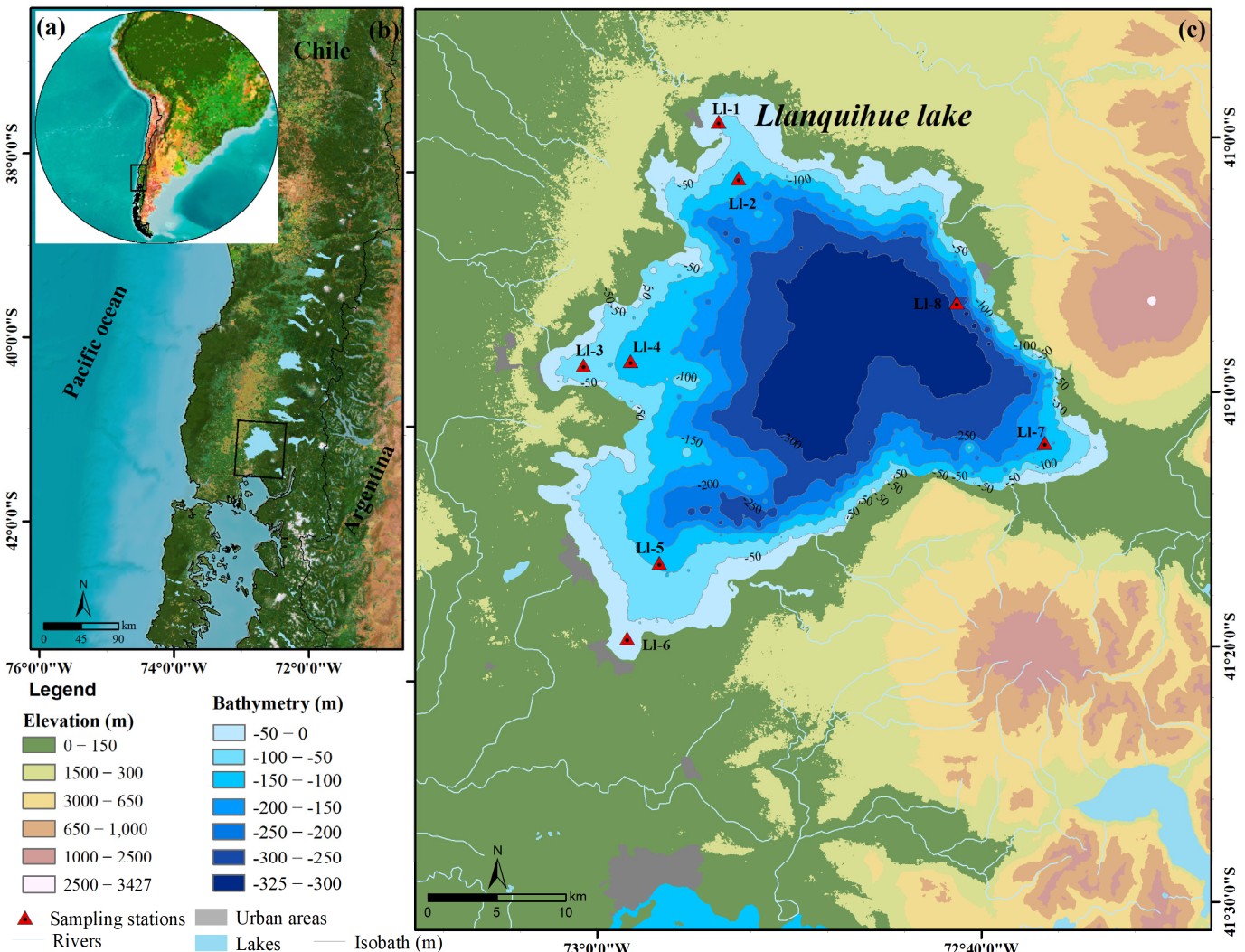

**Figure 1.** Location of the study area and sampling stations: (**a**) Chile in South America context, (**b**) location of Llanquihue lake in Chile, and (**c**) Llanquihue lake bathymetry and sampling stations (represented by red triangles).

## 2.2. Sample Collection

The Dirección General de Aguas de Chile (DGA for its acronym in Spanish) has been monitoring a group of lakes in Chile since 1986. Lake Llanquihue is within the selected group because it is the second-largest lake in the Chilean territory and because of its economic-social-cultural importance. The monitoring campaigns carried out in all seasons of the year consisted of sampling and in situ measurements of parameters at eight stations located in the lake (Ll1-Ll8, see Figure 1).

The collection of field data on physicochemical and biological parameters, as well as water samples for algae and pigment identification, is an essential part of water quality monitoring. The data collected will provide valuable information for understanding the

health of this water body and identifying potential environmental threats, such as the possible occurrence of algal blooms. The in-situ parameters selected for this study were Secchi disk depth (SD), chlorophyll-a (Chl-a) (Standard Methods N°10,200 H DGALGOCL1/2009), temperature (°C), total nitrogen (Nt) (Standard Methods N°4500-N C) and total phosphorus (Pt) (Standard Methods N°4500-P E). By following a standardized protocol and keeping detailed records, researchers and water managers can make informed decisions regarding water body management.

*2.3. Preprocessing of Landsat 8 Satellite Images*

Landsat-8 (L-8 OLI) images were used with a low percentage of clouds (less than 11%) covering the Llanquihue Lake (path/row: 233/89). L-8 is an Earth observation satellite of the Landsat project operated by the National Aeronautics and Spatial Administration (NASA) and the United States Geological Survey (USGS) [39,40]. It has two sensors, the OLI (Operational Land Imager) which provides nine bands in the visible, near-infrared, and shortwave spectra and covers from 0.433 μm to 1.390 μm, and the TIRS sensor (Thermal Infrared Sensor), which covers from 10.30 μm to 12.50 μm [41]. The 14 multispectral images used have a 30-m spatial resolution and were obtained from the USGS Earth Explorer (https://earthexplorer.usgs.gov/, accessed on 7 January 2023). The orthorectified and corrected images of the terrain of Collection 2 Level 1 were selected considering, the closeness to the sampling date and availability (see Table S1).

Considering a previous visual inspection through the Quality Assessment band (QA) and the Region of Interest (ROI), the images were atmospherically corrected in the ACO-LITE software (version 20211124.0) from https://github.com/acolite and accessed on 10 February 2023. ACOLITE is a generic processor that was developed specifically for marine, coastal, and inland waters, and brings together the atmospheric correction algorithms and software developed at RBINS for processing of images satellites applied to aquatic remote sensing [18,42,43]. ACOLITE uses a default atmospheric correction based on Dark Spectrum Fitting (DSF) [44–46] and Exponential Extrapolation (EXP) [43,47,48] algorithms.

From the resulting bands representing the surface-level reflectance (ρs) for L-8, the values were extracted in a matrix of 3 × 3 pixels per sampling point, according to [49]. Pixel values were extracted using ArcGIS software (ESRI's v. 10.8.2). Only data from cloud-free areas were used to have high-quality data and to avoid affecting the accuracy of the chlorophyll concentration estimate. These values were obtained from five multispectral bands: blue (B), green (G), red (R), near-infrared (NIR), and shortwave infrared (SWIR). In addition, the values of the Normalized Difference Vegetation Index (NDVI) and Floating Algal Index (FAI) were used. Both indexes are algorithms included in ACOLITE as part of the recovery of parameters derived from reflectance [50,51]. The limits of the lake were acquired from the DGA, and only the water body was considered for the analysis (DGA, 2023) [52].

Single bands are widely used to correlate with in situ data and estimate water quality parameters, such as chlorophyll [30,53], total suspended solids [54], turbidity [55], and temperature [56]. Surface reflectance values have shown good performance in these estimations and are even being used in artificial intelligence [57,58], including in algal bloom detection [24,32,55,59]. NDVI and FAI indices have been used in research related to chlorophyll and algal bloom estimation with good precision [18,60,61]. The NDVI is a commonly used indicator of vegetation photosynthetic activity and has been widely used in algae and chlorophyll extraction studies [62,63]. On the other hand, FAI is defined as a linear spread of reflectivity in the near-infrared, red, and shortwave infrared regions and can be applied to monitor algal blooms. The observation results of this algorithm provide a high accuracy [61,63,64].

*2.4. Prediction Using Statistical and Deep Learning Models*

2.4.1. SARIMAX

The SARIMAX (Seasonal Autoregressive Integrated Moving Average with Exogenous Variables) model is a type of time series model. It is an extension of the ARIMA (Autore-

gressive Integrated Moving Average) model that incorporates both seasonal and exogenous components [65]. The mathematical model is defined by Equation (1):

$$y_t = \beta_t x_t + u_t$$
$$\varphi_p(L)\tilde{\phi}_p(L^s)\Delta^d\Delta_s^D u_t = A(t) + \theta_q(L)\tilde{\theta}_Q(L^s)\zeta$$

(1)

where $\beta$ in the first part of the formula represents external variables. The model is similar to the SARIMA model, with the following hyperparameters [66]:

p represents the order for the Autoregressive part (AR)
q represents the order for the moving average part (MA)
I represents the differencing order
P represents the seasonal AR order
Q represents the seasonal MA order
D represents the seasonal differencing
s represents the seasonal coefficients

The complete data pipeline used is shown in Figure 2. Raw data were obtained for each station (Figure 1). Subsequently, data were resampled at monthly intervals. Autocorrelation (ACF) and Partial Autocorrelation (PACF) functions were computed. Additionally, the Dickey Fuller test [67] was employed to examine whether the chlorophyll-a time series exhibited the characteristics of white noise. Subsequently, a seasonal order was applied, and, if necessary, a 1-step differentiation was performed. The optimal hyperparameters were determined through calibration using the pmdarima library (https://github.com/alkaline-ml/pmdarima, (accessed on 20 February 2023)), and the range of values for calibration was established based on the ACF and PACF results. Finally, the most suitable model was selected based on Akaike Information Criteria (AIC) and Bayesian Information Criteria (BIC).

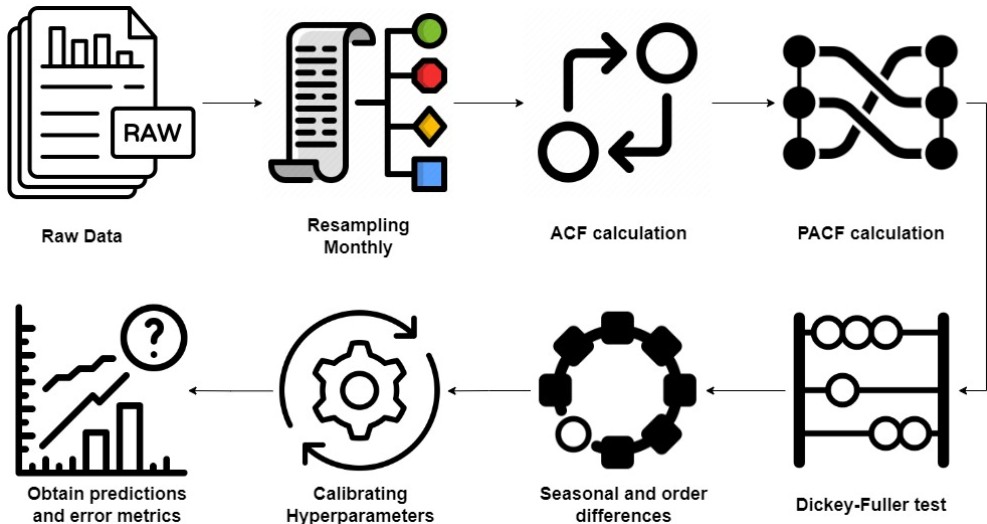

**Figure 2.** Flowchart of the integrated autoregressive moving average ARIMA model.

2.4.2. Long Short-Term Memory (LSTM)

Subsequently, Long short-term memory (LSTM) was used, which is a variant of a Recurrent neural network (RNN) proposed by Hochreiter and Schmidhuber in 1997 [68]. This algorithm solves the long-term dependency problem in RNNs by introducing memory (C) and an appropriate gate structure.

The LSTM cell (Figure 3) has four gates: input ($i$), forget ($f$), control ($c$) and output gates ($o$). The input gate determines the information that can be inserted and transferred to the cell:

$$i_t = \sigma(W_i \cdot [h_{t-1}, x_t] + b_i)$$

(2)

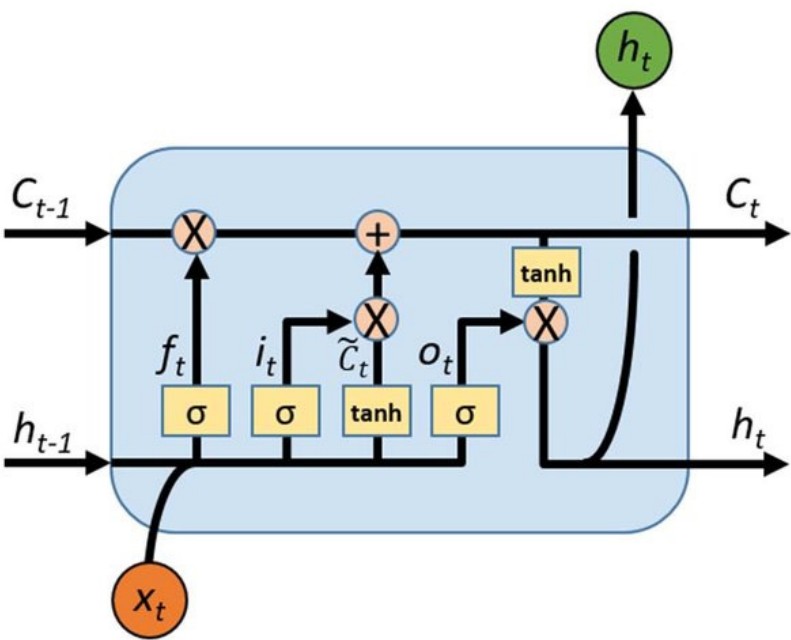

**Figure 3.** Structure of a Long Short-Term Memory (LSTM) cell. Adapted from [69].

The forget gate decides which information from the input is important from previous memory with:

$$f_t = \sigma\left(W_f \cdot [h_{t-1}, x_t] + b_f\right) \tag{3}$$

The control gate stabilizes the update in the cell state from $C_{t-1}$ to $C_t$ using Equations (4) and (5):

$$\tilde{C}_t = \tanh(W_C \cdot [h_{t-1}, x_t] + b_c) \tag{4}$$

$$C_t = f_t \times C_{t-1} + \tilde{C}_t \tag{5}$$

The output gate generates the output updating the hidden vector $h_{t-1}$ with Equations (6) and (7):

$$o_t = \sigma_t \times \tanh(C_t) \tag{6}$$

$$h_t = o_t \times \tanh(C_t) \tag{7}$$

where $\sigma$ is the activation function, $W$ corresponds to the weight matrices calibrated during the training process, tanh is used to scale values in the range of $-1$ to 1, and b represents the bias in each step. During the training process, a lag of 9 is constructed from the input variables. An LSTM layer with a variable number of cells ranging from 30 to 50 was used based on the size and complexity of the dataset. Furthermore, a Dense layer is employed as the output layer to facilitate accurate predictions. The latter topology is considered to be a common configuration in the LSTM algorithm [69]. The complete training structure is described in Figure 4.

2.4.3. Recurrent Neural Networks (RNN)

We incorporated the architecture of Recurrent Neural Networks (RNN) to assess the performance of the LSTM models relative to their predecessor. RNN are a class of neural networks that are suitable for sequential data such as time series [68]. This algorithm is just a feed-forward neural network that unfolds over time (Figure 5). At each time step, the network produces an intermediate output $o_t$ and maintains an internal state $s_t$; therefore, $x_t$ creates the sequential input that is given to the network following Equations (8)–(11):

$$a_t = b + Ws_{t-1} + Ux_t \tag{8}$$

$$s_t = \tanh(a_t) \tag{9}$$

$$o_t = c + Vs_t \tag{10}$$

$$y_t = softmax(o_t) \tag{11}$$

where *U, V* and *W* represent matrices with the parameters of the model learned by standard propagation, b represents the bias, and $y_t$ represents the final output and linear represents the activation function which in this case is the identity transformation defined in Equation (12) as previously described by Rumelhart et al. (1986) [70].

$$linear(o_t) = o_t$$

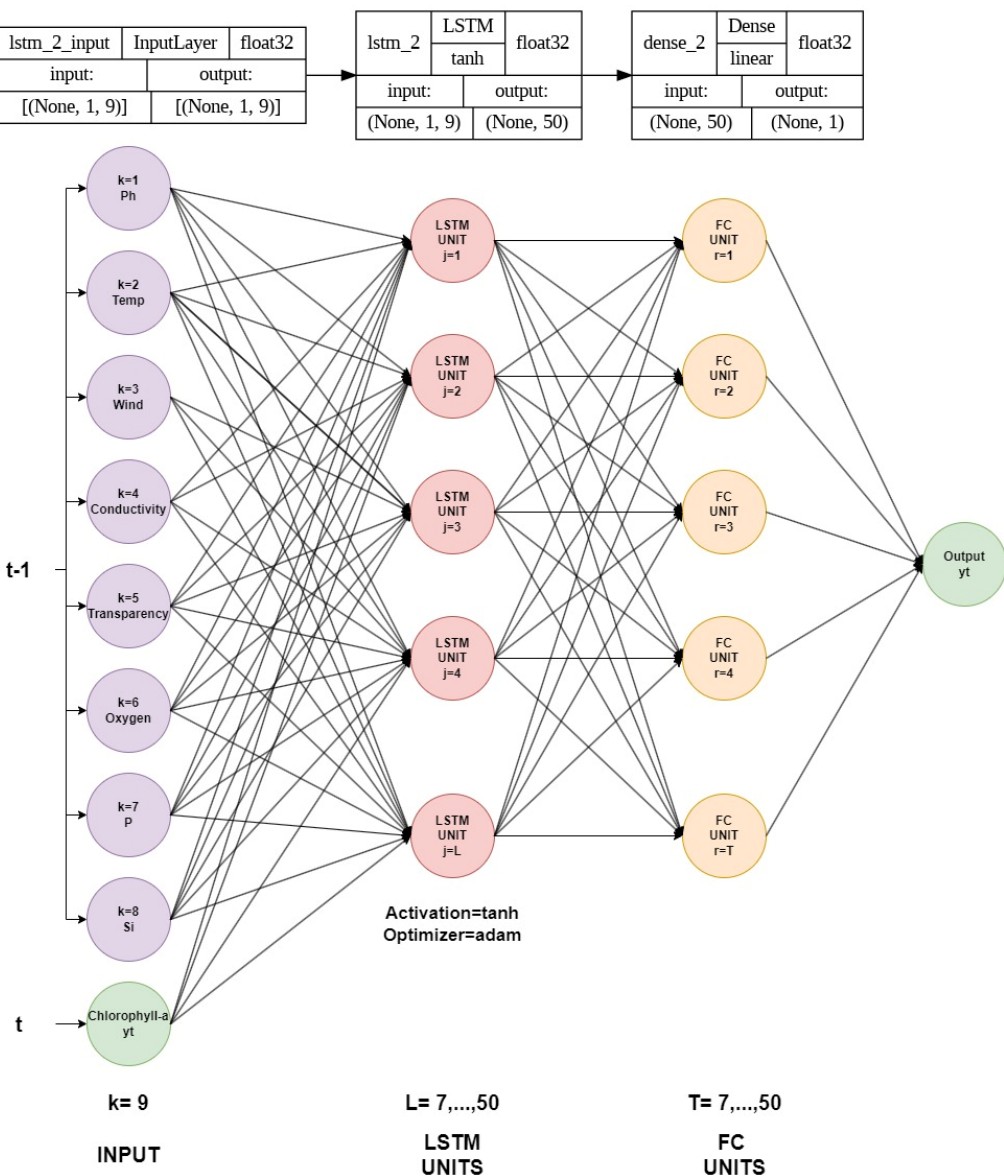

**Figure 4.** Schematic representation of LSTM structure used for the training in the LI-7 station. k represents the number of variables (including independent and dependent outcome) depending on the case it could be 9 (A),12 (B) or 14 (C). L represent the number of LSTM units for the training with values between 7 and 50 depending on the data complexity.

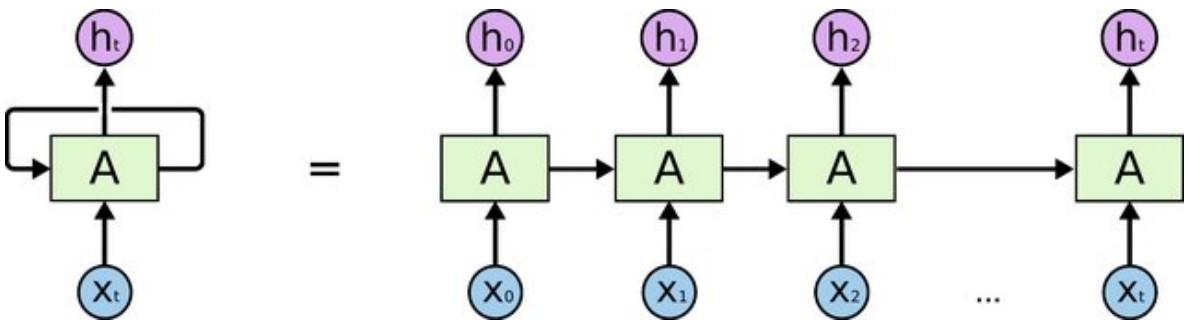

**Figure 5.** RNN network architecture with a loop (adapted from [71]).

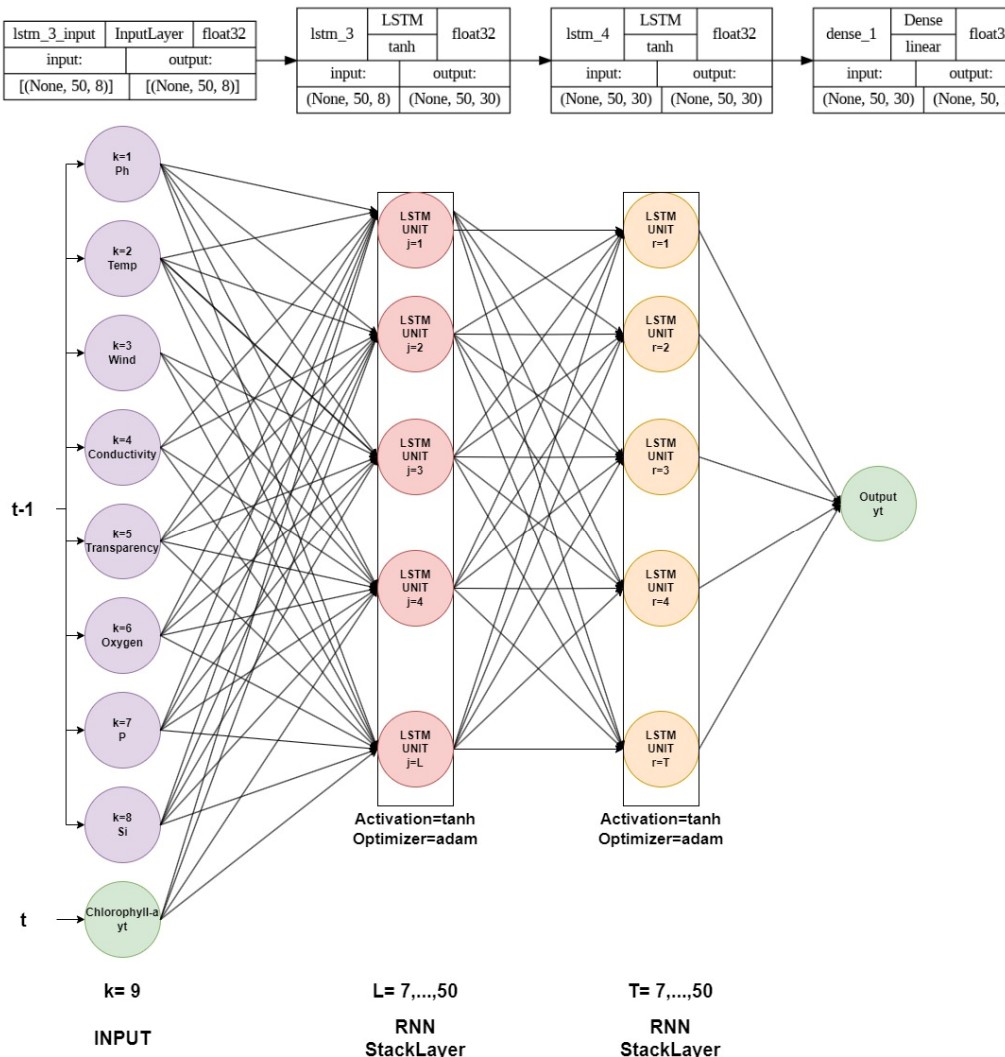

**Figure 6.** Schematic representation of RNN structure used for the training in the LI-7 station. k represents the number of variables (including independent and dependent outcome) depending on the case it could be 9 (A),12 (B) or 14 (C). L and T represent the number of LSTM units for the training with values between 10 and 30 depending on the data complexity.

To train this network, we use a sliding window technique to generate training data for a time-series prediction model using a window of size lookback between 15 and 30 from the input time series. The corresponding target value (chlorophyll-a) was set as the next time step after the input sequence. In this way, the model can learn from the input-output pairs and make predictions based on the observed patterns in the training data.

Therefore, the data enter the RNN topology passing through a number of units between 16 and 32, depending on the amount of data and complexity, and finally, a dense layer produces the predictions. The optimizer and loss metric used were Adam and mean squared error, respectively. This configuration has been widely used in previous studies on time-series predictions [68]. The complete training structure is described in Figure 6.

To test the three models described above, cases A, B, and C are defined as follows:

Case A (Measurement Data): In the first case, we included the real variables measured in the monitoring campaigns for the four seasons of the year and in the eight stations of the lake.

Case B (Measurement and Meteorological Data): In addition to the actual variables, we included meteorological data as conditioning variables that can influence the autochthonous processes of the lake.

Case C (Measurement, Meteorological Data and Satellite Data): In this case, we include bands and indices from the L-8 satellite image processing.

### 2.5. Statistical Validation

To analyze the performance of the models defined in Section 2.4, several metrics were used, such as Mean Squared Error (MSE) as described in [72], Root Mean Squared Error (RMSE) described in [30], Mean Absolute Error (MAE), Maximum Error describe in [73] and $R^2$ described in [74] following a similar approach as the one applied in [35]. Thus, it helps in understanding the accuracy, precision, and potential limitations of estimating chlorophyll-a. Number of samples for train and test are described in Table 1. Sequential splitting with a 70/30% rule was used to calculate the different error metrics. In this method the time series are separated as follows, the earlier (later) is used to train (validate) each model across the different stations (Figure 1) and have been demonstrated good performance to assess time-series performance in deep learning models [75,76].

**Table 1.** Number of samples used for (train/test) in the train and validation phases for each station and case analyzed.

| Case | LI-1 | LI-2 | LI-3 | LI-4 | LI-5 | LI-6 | LI-7 | LI-8 |
|------|------|------|------|------|------|------|------|------|
| Case A | 238/59 | 77/19 | 238/59 | 92/23 | 238/59 | 80/35 | 332/83 | 67/29 |
| Case B | - | 36/8 | - | 36/10 | - | 36/10 | 36/12 | 36/8 |
| Case C | - | 36/8 | - | 36/10 | - | 36/10 | 36/12 | 36/8 |

On the other hand, we used a method called Garson's weighting to assign importance or weights to the input variables in neural networks. This method provides a measure of the relative contribution of each predictor variable explaining the variation in the dependent variable [77]. To obtain weights the method uses the magnitude and direction of the coefficients obtained from the model. Variables with larger absolute coefficients are considered to have higher importance. In this way, the most important variables in the model were obtained for each case of study.

## 3. Results

### 3.1. Limnological Behavior and Meteorological Data

The water quality and trophic level of a lake primarily depend on nutrient inputs (Nitrogen and Phosphorus) from the watershed. This is why they are selected in conjunction with the transparency, the temperature, and the study variable chlorophyll a. All these parameters influence the spatio-temporal distribution of algae. Figure 7 shows the behavior of limnological variables associated with algal blooms. The boxplot shows the distribution of the numerical dataset of the variables indicated by five key statistics: minimum value, first quartile (Q1), median (Q2), third quartile (Q3), and maximum value (Q4).

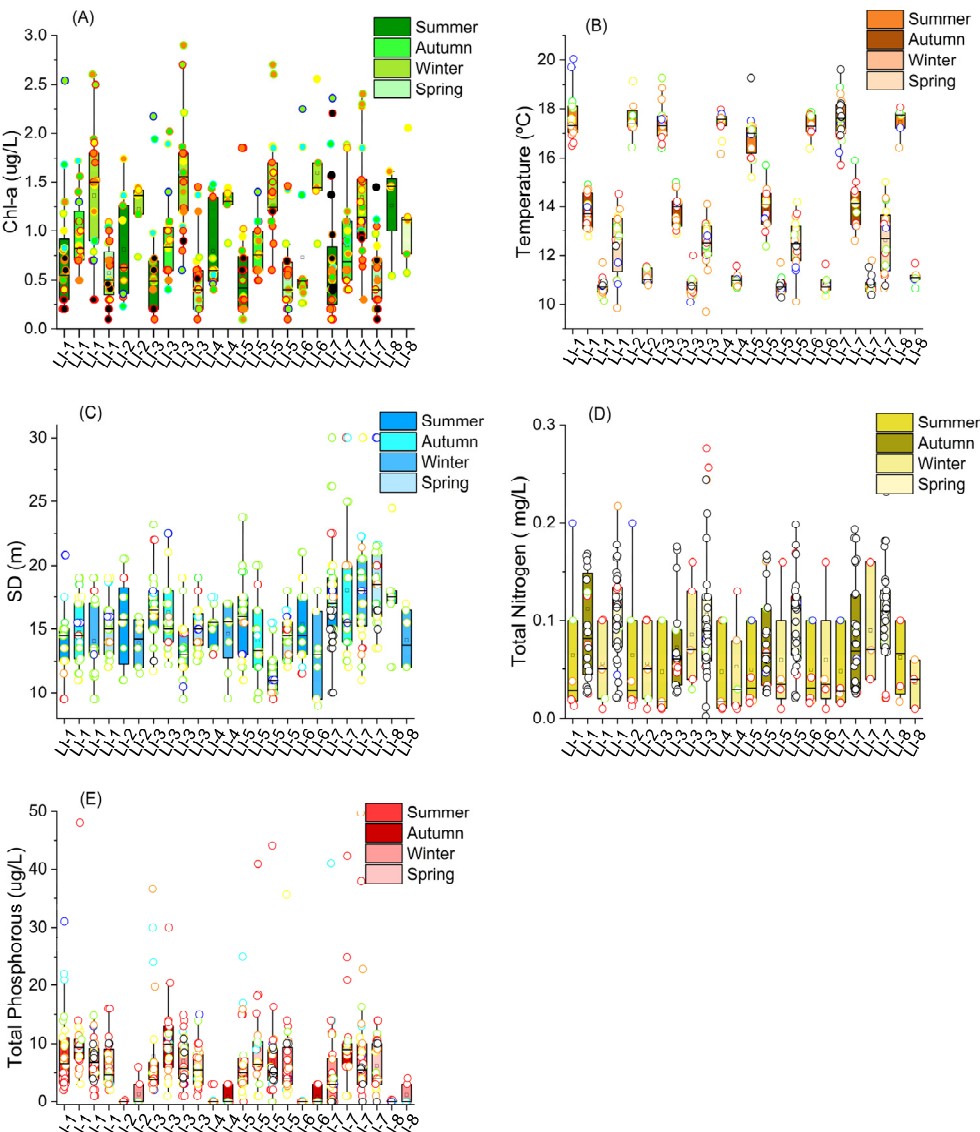

**Figure 7.** Boxplot of limnological parameters in Llanquihue Lake over 1986–2021: (**A**) chlorophyll-a (µg/L), (**B**) temperature (°C), (**C**) SD (m), (**D**) total nitrogen (mg/L), (**E**) total phosphorous (µg/L).

Chlorophyll-a values vary widely depending on location, season, and environmental conditions (see Table 2). Chl-a in Lake Llanquihue ranged from a minimum of Q1 = 0.50 µg/L to a maximum of Q4 = 2.90 µg/L, for all other statistics see Table S2. Water temperature varied according to seasonality, an expected result with a winter minimum of 1.6 °C and a summer maximum of 20 °C, the temperature minima and maxima coinciding with the Chl-a minima and maxima. Nitrogen and total phosphorus were analyzed in the lake system. The recorded values of nitrogen were low between 0.003 and 0.6 mg/L, while phosphorous was found in higher concentrations between 1.0 and 56.0 µg/L. Transparency is generally high in Llanquihue Lake during most of the year and for all lake seasons with a maximum of 30 m and a minimum of 6 m in winter, which may be attributed to turbidity related to precipitation events and strong winds during this time of year in the Southern Hemisphere.

**Table 2.** Meteorological variable for Llanquihue Lake.

| Months | Temperature (°C) | Wind Speed (m/s) | Relative Humidity (%) | Cloud Cover (%) | Accumulated Precipitation (mm) | Photosynthetic Active Radiation (mmol/m$^2$) |
|---|---|---|---|---|---|---|
| January | 16.43 | 3.90 | 62.50 | 0.50 | 60.07 | 63,581.1 |
| February | 18.42 | 3.30 | 75.50 | 0.33 | 47.67 | 56,124.7 |
| March | 17.77 | 2.70 | 65.20 | 0.55 | 75.00 | 33,109.3 |
| April | 14.89 | 3.80 | 89.70 | 0.70 | 121.09 | 19,872.1 |
| May | 12.15 | 4.10 | 76.20 | 0.80 | 184.7 | 15,883.9 |
| June | 9.40 | 3.90 | 71.80 | 1.00 | 239.60 | 11,225.7 |
| July | 8.64 | 4.10 | 78.30 | 0.90 | 205.40 | 10,393.8 |
| August | 12.05 | 2.90 | 73.30 | 0.90 | 207.90 | 17,965.5 |
| September | 13.11 | 2.70 | 76.15 | 0.72 | 110.70 | 30,330.5 |
| October | 14.00 | 4.07 | 62.80 | 0.69 | 105.60 | 42,777.1 |
| November | 14.89 | 3.90 | 66.01 | 0.55 | 80.90 | 54,885.1 |
| December | 15.66 | 4.20 | 92.70 | 0.45 | 57.10 | 62,428.7 |

### 3.2. Results and Validation of Statistical and Deep Learning Models Cases

### 3.2.1. Case A (Measurement Data)

Figure 8 shows the behavior of the estimated chlorophyll-a at the eight-sampling stations during the study period. In each case, we modeled Chl-a for the three models using SARIMAX, LSTM and RNN.

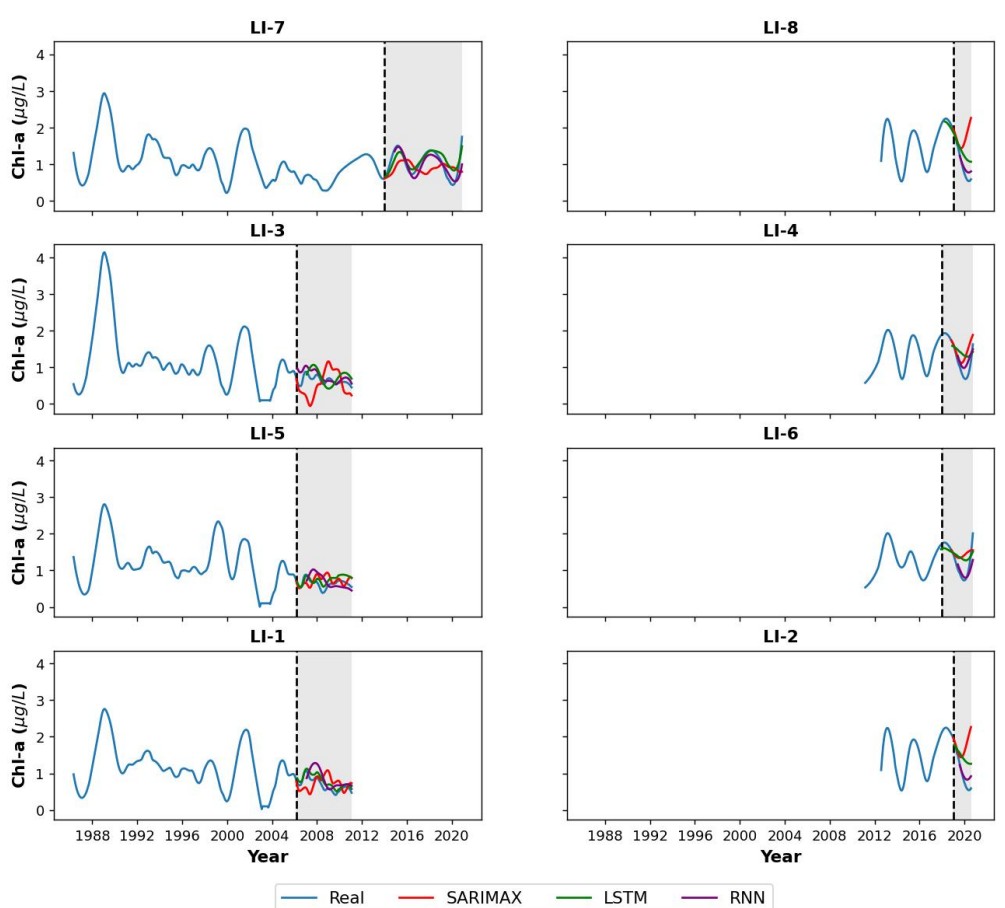

**Figure 8.** Chlorophyll-a estimation of case A in the eight-sampling stations of Llanquihue Lake during 1988–2020. The shaded regions represent observations for the selection of the test.

From the results, we can observe that in most of the stations, the models offer a good retrieval of the temporal variations, except for LI-3, LI-2, and LI-8 when the SARIMAX model

is used (red line). In addition, the SARIMAX model exhibits higher MSE, RMSE, MaxError, and MAE ($R^2 < 0.915$) values compared to the other models (RNN and LSTM) (Table 3).

**Table 3.** Validation metrics for all the stations and models considered in Case A (Section 2.4).

| | | Case A | | | | | | | |
|---|---|---|---|---|---|---|---|---|---|
| Station | Statistic | LI-1 | LI-2 | LI-3 | LI-4 | LI-5 | LI-6 | LI-7 | LI-8 |
| SARIMAX | MSE ($\mu$g/L)$^2$ | 0.083 | 0.787 | 0.160 | 0.173 | 0.043 | 0.190 | 0.139 | 0.787 |
| | RMSE ($\mu$g/L) | 0.288 | 0.887 | 0.400 | 0.415 | 0.206 | 0.436 | 0.372 | 0.887 |
| | MaxError ($\mu$g/L) | 0.505 | 1.676 | 0.783 | 0.726 | 0.459 | 0.700 | 0.955 | 1.676 |
| | MAE ($\mu$g/L) | 0.244 | 0.660 | 0.350 | 0.324 | 0.159 | 0.366 | 0.322 | 0.660 |
| | $R^2$ | 0.892 | 0.724 | 0.857 | 0.864 | 0.915 | 0.685 | 0.793 | 0.795 |
| LSTM | MSE ($\mu$g/L)$^2$ | 0.014 | 0.260 | 0.029 | 0.166 | 0.020 | 0.101 | 0.039 | 0.098 |
| | RMSE ($\mu$g/L) | 0.116 | 0.510 | 0.169 | 0.407 | 0.142 | 0.317 | 0.199 | 0.314 |
| | MaxError ($\mu$g/L) | 0.194 | 0.730 | 0.389 | 0.625 | 0.244 | 0.563 | 0.423 | 0.552 |
| | MAE ($\mu$g/L) | 0.106 | 0.442 | 0.136 | 0.348 | 0.121 | 0.263 | 0.152 | 0.247 |
| | $R^2$ | 0.912 | 0.932 | 0.896 | 0.912 | 0.934 | 0.854 | 0.893 | 0.936 |
| RNN | MSE ($\mu$g/L)$^2$) | 0.056 | 0.045 | 0.046 | 0.066 | 0.0509 | 0.068 | 0.030 | 0.028 |
| | RMSE ($\mu$g/L) | 0.236 | 0.212 | 0.214 | 0.257 | 0.225 | 0.260 | 0.174 | 0.167 |
| | MaxError ($\mu$g/L) | 0.447 | 0.331 | 0.383 | 0.379 | 0.451 | 0.724 | 0.754 | 0.238 |
| | MAE ($\mu$g/L) | 0.196 | 0.176 | 0.192 | 0.237 | 0.191 | 0.183 | 0.127 | 0.146 |
| | $R^2$ | 0.901 | 0.915 | 0.876 | 0.893 | 0.926 | 0.827 | 0.843 | 0.648 |

Furthermore, the LSTM model generally performs well across most stations, with lower values for MSE ($<0.260$ ($\mu$g/L)$^2$), RMSE ($<0.510$ ug/L), MaxError ($<0.730$ $\mu$g/L), and MAE ($<0.442$ $\mu$g/L) compared to SARIMAX, with higher values at LI-2 station. Additionally, the $R^2$ values for LSTM were consistently high, indicating a good fit to the data. In addition, the RNN model shows similar performance to LSTM, with relatively low MSE ($<0.068$ ($\mu$g/L)$^2$), RMSE ($<0.260$ ug/L), MaxError ($<0.751$ $\mu$g/L), and MAE ($<0.283$ $\mu$g/L) values, and the $R^2$ values were also consistently high ($>0.827$) (Table 3).

### 3.2.2. Case B (Measurement and Meteorological Data)

Figure 9 shows the results for Case B. In all cases, the estimation of the chlorophyll-a values was similar. Sampling stations 1, 3 and 5 did not have sufficient data to estimate the variable with the models.

Similarly, for Case-A, the SARIMAX model showed a moderate performance compared to the RNN and LSTM models with the MSE, RMSE, MaxError, and MAE metrics relatively lower (higher) for station LI-6 (LI-8) compared to the others. In addition, the $R^2$ values for SARIMAX are generally approximately equal to 0.8 (Table 4).

In contrast, the LSTM model generally performs well across all stations with relatively low MSE ($<0.029$ ($\mu$g/L)$^2$), RMSE ($<0.172$ ug/L), MaxError ($<0.175$ $\mu$g/L), and MAE ($<0.172$ $\mu$g/L) values. The $R^2$ values are lower compared to Case A but considering the smaller amount of data used for the training, there is still a good performance ($>0.80$). Besides the RNN model showed a similar performance to LSTM, with low MSE, RMSE, MaxError, and MAE values across all stations, and $R^2$ values above 0.80 (Table 4).

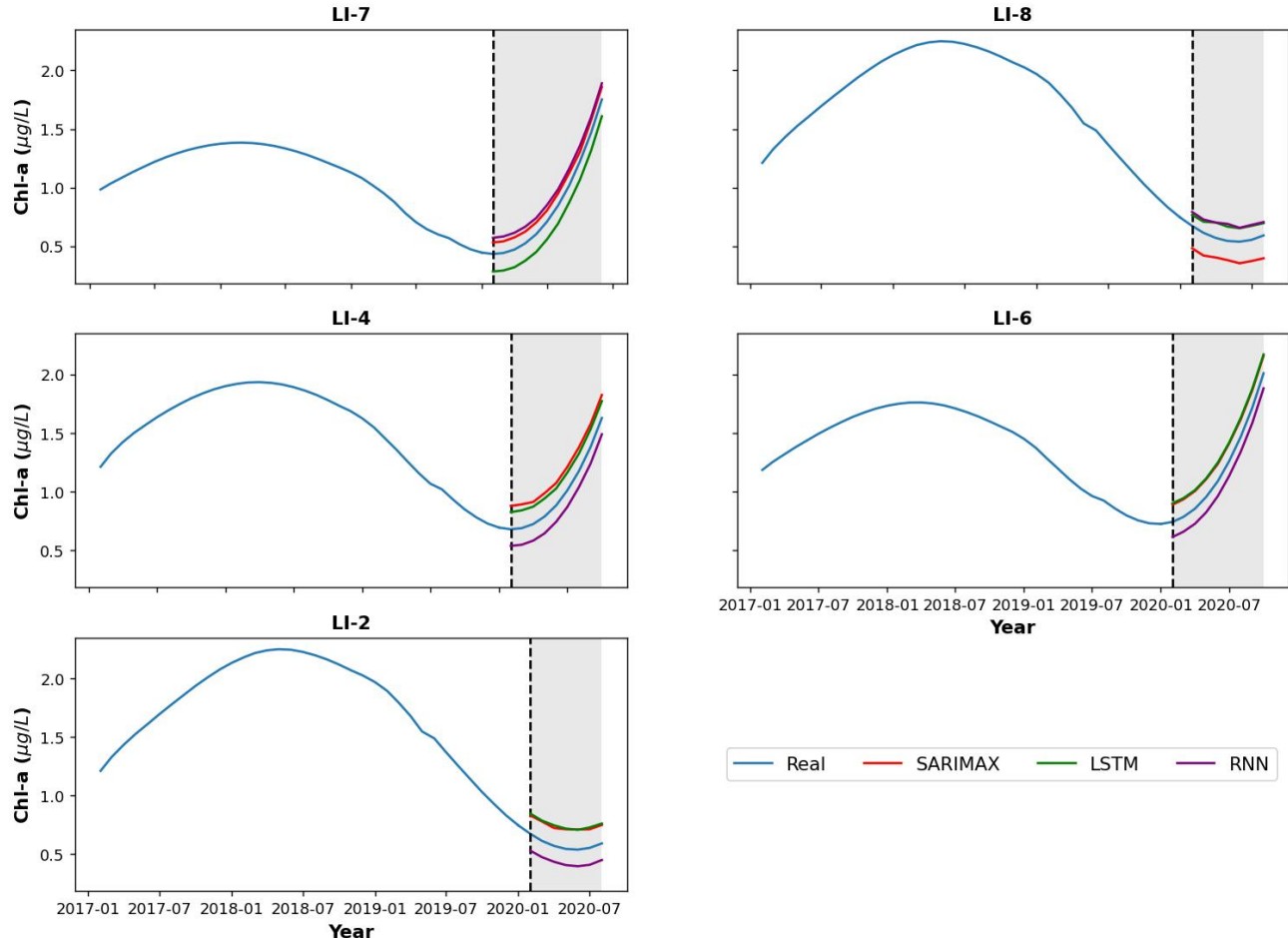

**Figure 9.** Chlorophyll-a estimation of Case B at the five sampling stations with data from Llanquihue Lake during 1988–2020. The shaded regions represent observations for the selection of the test.

**Table 4.** Validation metrics for all the stations and models considered in Case B (Section 2.4).

| | Case B | | | | | |
|---|---|---|---|---|---|---|
| **Station** | **Statistic** | **LI-2** | **LI-4** | **LI-6** | **LI-7** | **LI-8** |
| SARIMAX | MSE $(ug/L)^2$ | 0.026 | 0.039 | 0.023 | 0.010 | 0.033 |
| | RMSE (ug/L) | 0.162 | 0.197 | 0.150 | 0.101 | 0.183 |
| | MaxError (ug/L) | 0.172 | 0.203 | 0.151 | 0.108 | 1.195 |
| | MAE (ug/L) | 0.162 | 0.197 | 0.149 | 0.102 | 0.182 |
| | $R^2$ | 0.795 | 0.781 | 0.797 | 0.776 | 0.812 |
| LSTM | MSE $(ug/L)^2$ | 0.029 | 0.022 | 0.025 | 0.026 | 0.013 |
| | RMSE (ug/L) | 0.172 | 0.149 | 0.159 | 0.150 | 0.112 |
| | MaxError (ug/L) | 0.175 | 0.154 | 0.161 | 0.157 | 0.131 |
| | MAE (ug/L) | 0.172 | 0.149 | 0.159 | 0.150 | 0.111 |
| | $R^2$ | 0.816 | 0.842 | 0.837 | 0.821 | 0.834 |
| RNN | MSE $(ug/L)^2$ | 0.019 | 0.020 | 0.016 | 0.019 | 0.016 |
| | RMSE (ug/L) | 0.141 | 0.141 | 0.129 | 0.139 | 0.125 |
| | MaxError (ug/L) | 0.144 | 0.144 | 0.131 | 0.144 | 0.146 |
| | MAE (ug/L) | 0.140 | 0.141 | 0.129 | 0.139 | 0.124 |
| | $R^2$ | 0.805 | 0.840 | 0.830 | 0.824 | 0.806 |

### 3.2.3. Case C (Measurement, Meteorological Data and Satellite Data)

Figure 10 shows Case C, which is the most general and complete compared to the two cases presented before because it integrates all types of data available: measurement in situ data, meteorological data, and satellite data.

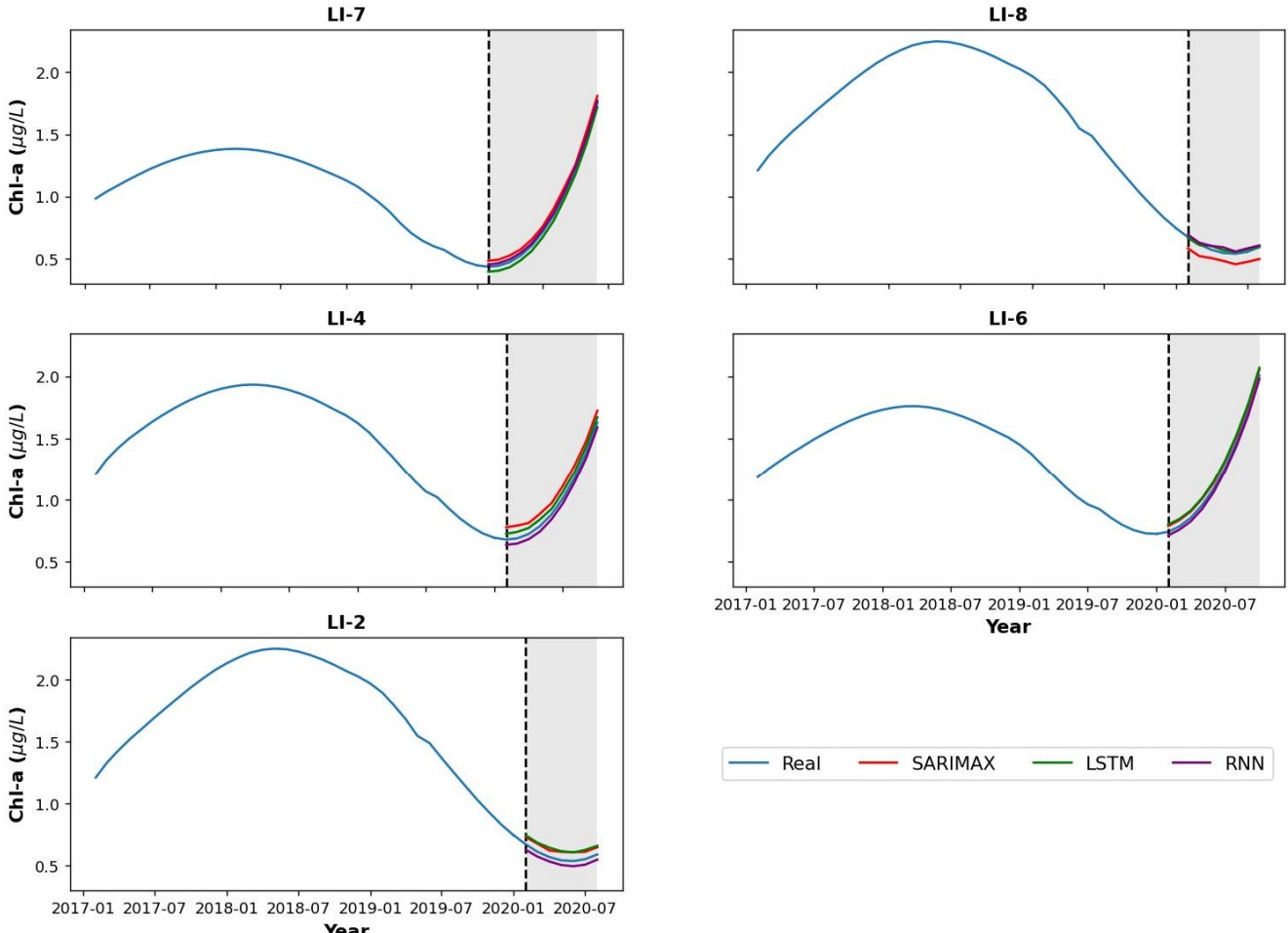

**Figure 10.** Chlorophyll-a estimation of case C in the five sampling stations with data from Llanquihue Lake during 1988–2020. The shaded regions represent observations for the selection of the test.

When comparing the results between Table in Case C vs. Case B, we observe that all the models' performance is better in Case C, which is evident with lower values for MSE (<0.009 μg/L)$^2$), RMSE (<0.097 ug/L), MaxError (<0.103 ug/L), and MAE (<0.097 ug/L) and higher $R^2$ values (<0.81); however, LSTM and RNN showed better performance against SARIMAX (Table 5). This suggests that incorporating a larger set of chlorophyll-a-related variables both directly and indirectly enhances the predictive capacity of the algorithms.

### 3.3. Feature Importance

Figures 11–13 also show, using Garson's weighting method [78], the relative importance or contribution of the independent variables (predictor variables) in explaining the variance of the dependent variable (outcome variable) for each case used.

**Table 5.** Validation metrics for all the stations and models considered in Case C (Section 2.4).

| | | Case C | | | | |
|---|---|---|---|---|---|---|
| SARIMAX | MSE ($\mu$g/L)$^2$ | 0.003 | 0.009 | 0.002 | 0.002 | 0.006 |
| | RMSE (ug/L) | 0.062 | 0.097 | 0.050 | 0.050 | 0.083 |
| | MaxError (ug/L) | 0.07 | 0.103 | 0.051 | 0.057 | 0.095 |
| | MAE (ug/L) | 0.062 | 0.097 | 0.049 | 0.050 | 0.082 |
| | $R^2$ | 0.804 | 0.807 | 0.812 | 0.832 | 0.796 |
| LSTM | MSE ($\mu$g/L)$^2$ | 0.001 | 0.002 | 0.003 | 0.002 | 0.001 |
| | RMSE (ug/L) | 0.072 | 0.049 | 0.060 | 0.040 | 0.018 |
| | MaxError (ug/L) | 0.075 | 0.054 | 0.061 | 0.046 | 0.031 |
| | MAE (ug/L) | 0.072 | 0.049 | 0.059 | 0.040 | 0.015 |
| | $R^2$ | 0.857 | 0.864 | 0.896 | 0.877 | 0.843 |
| RNN | MSE ($\mu$g/L)$^2$ | 0.001 | 0.002 | 0.001 | 0.001 | 0.001 |
| | RMSE (ug/L) | 0.041 | 0.041 | 0.029 | 0.019 | 0.018 |
| | MaxError (ug/L) | 0.044 | 0.044 | 0.031 | 0.024 | 0.031 |
| | MAE (ug/L) | 0.045 | 0.041 | 0.029 | 0.019 | 0.015 |
| | $R^2$ | 0.843 | 0.832 | 0.815 | 0.807 | 0.795 |

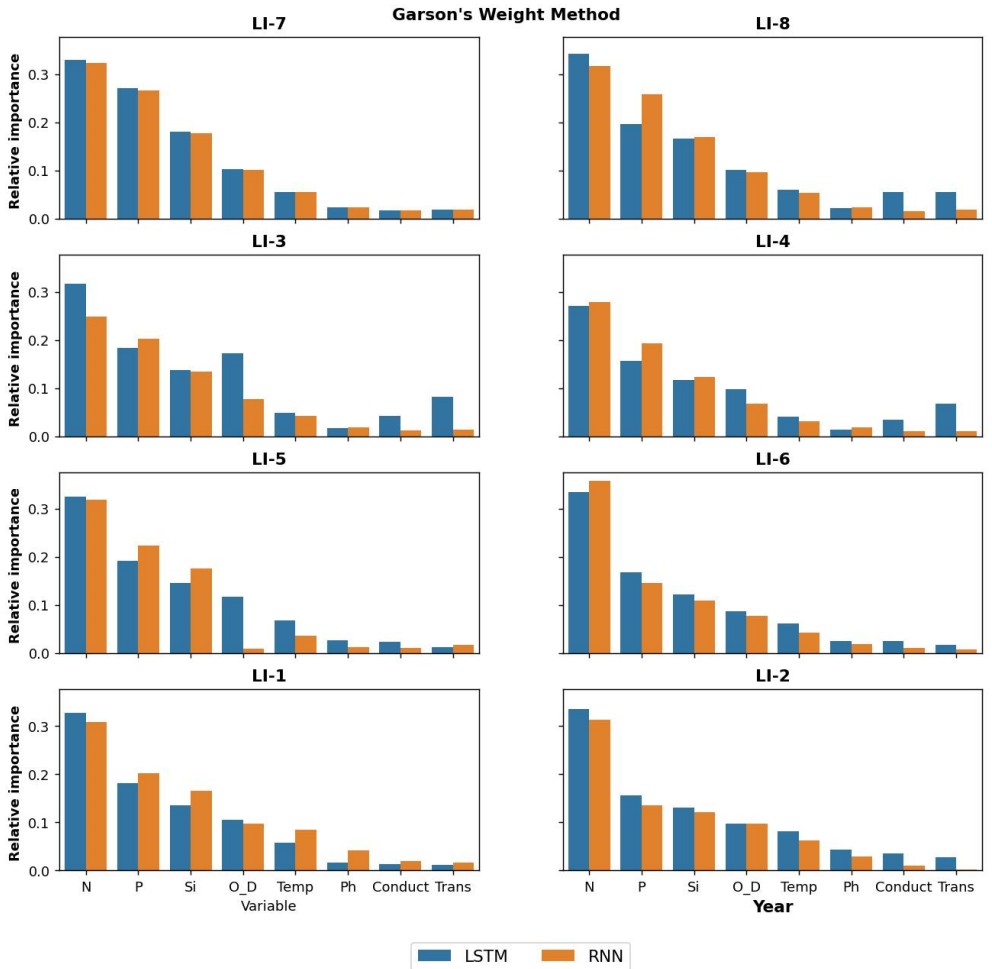

**Figure 11.** Relative importance using Garson's weighting method Case A for the eight-sampling stations. N is Nitrogen, P is Phosphorus, Si Silicon, O_D Dissolved Oxygen, Temp temperature, Conduct Conductivity, and Trans Transparency.

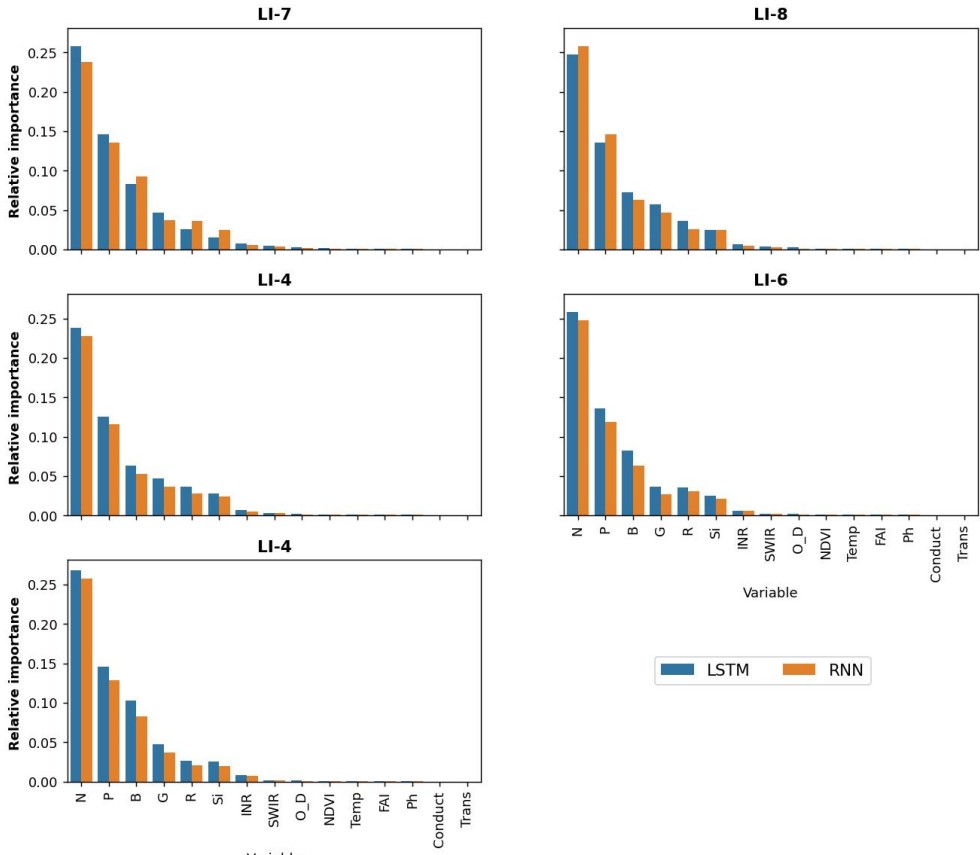

**Figure 12.** Relative importance through Garson's weighting method for Case B at five-sampling stations. N is Nitrogen, P is Phosphorus, Si Silicon, O_D Dissolved Oxygen, Temp temperature, Conduct Conductivity, Trans *Transparency*, B blue band, G green band, R red band, INR near infrared band, SWIR shortwave infrared band, NDVI Normalized Difference Vegetation Index, and FAI, Floating Algae Index.

Results showed that Nitrogen (N), Phosphorus (P), and Silica (Si) present the highest feature importance values (ranging from 0.115 to 0.336) in predicting Chlorophyll-a across all stations in case A. Subsequently, Dissolved Oxygen (O_D) and Temperature (Temp) showed relative importance ranging from 0.053 to 0.157, with O_D being more significant than Temp. Conversely, pH, Conductivity (Conduct), and Transparency (SD) exhibited relatively lower importance, all having values below 0.1.

For Case B (Figure 12), we can see that adding satellite images modifies the order of importance of some of the variables. Similarly, N and P variables showed higher relative importance, with contributions of >0.15. However, the B, G, and R bands were more important (ranging from 0.033 to 0.046) with respect to the Si variable, with contributions < 0.038.

Finally, in Case C (Figure 13), a pattern comparable to that of Case B was observed. The variables N and P remained the most crucial factors in the predictions (relative importance above 0.110), followed by the B, G, and R bands, and subsequently the Si variable. Therefore, the variables INR, SWIR, and O_D exhibit the most significant contributions, and the remaining variables have a negligible importance of <0.035.

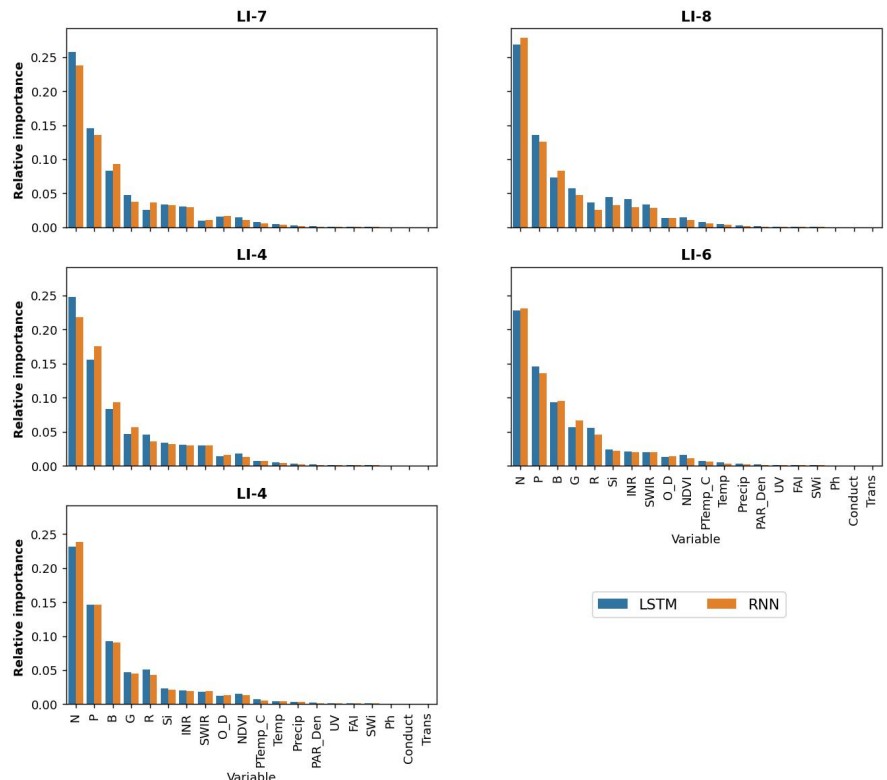

**Figure 13.** Relative importance using Garson's weighting method for Case C. In five-sampling station. N is Nitrogen, P is Phosphorus, Si Silicon, O_D Dissolved Oxygen, Temp temperature, Conduct Conductivity, Trans *Transparency*, B blue band, G green band, R red band, INR near infrared band, SWIR shortwave infrared band, NDVI Normalized Difference Vegetation Index, and FAI, Floating Algae Index.

## 4. Discussion

It is important to monitor the behavior of the lakes since they are sentinels or indicators of climate change [79]. Human activities accelerate the eutrophication processes of these southern freshwater ecosystems. In this study, using combined remote sensing and machine learning techniques, models were created to estimate water quality parameters such as chlorophyll-a in a lake in southern Chile. Llanquihue Lake is the lake body most vulnerable to contamination among the lakes of the Araucanian Lake district, owing to its slower water mass renewal time than the rest (estimated to be 74 years), in addition to the intense use of its shores [35]. It is the only lake in Chile, with four municipalities located on its shores, all of which are the main capitals: Puerto Octay, Frutillar, Llanquihue and Puerto Varas. Currently, Llanquihue Lake, despite maintaining the trophic characteristics of oligotrophic lakes, the values of key water quality parameters such as chlorophyll-a and nutrients have increased in Llanquihue Lake, and their trophic level can change in a shorter time than the natural succession process of aquatic systems. In a previous study, we calibrated and validated a set of nine artificial intelligence algorithms over a period longer than 30 years to estimate the chlorophyll-a variable at different points of the lake [35].

In the present work, we aimed to add, in addition to the historical data from the lake monitoring campaigns conducted by the General Water Directorate of Chile, data from the Meteorological Directorate of Chile and data from Landsat-8 satellite images. Excellent and accurate results were obtained for each season of the year in this lake. Model validation metrics showed that all three models effectively predicted chlorophyll as an indicator of the presence of algae in this water body. Coefficient of determination values ranging from 0.64 to 0.93 were obtained, with the LSTM model showing the best statistics in any of the cases tested, and similar results were obtained in [80] when predicting chlorophyll using the LSTM model. The LSTM model generally performs well across most stations, with lower

values for MSE ($<0.260$ $(\mu g/L)^2$), RMSE ($<0.510$ ug/L), MaxError ($<0.730$ µg/L), and MAE ($<0.442$ µg/L) compared to SARIMAX, with higher values at LI-2 station. Additionally, the $R^2$ values for the LSTM were consistently high, indicating a good fit to the data. In addition, the RNN model shows similar performance as LSTM, with relatively low MSE ($<0.068$ $(\mu g/L)^2$), RMSE ($<0.260$ ug/L), MaxError ($<0.751$ µg/L), and MAE ($<0.283$ µg/L) values, and the $R^2$ values were also consistently high ($>0.827$). When comparing the results between Case C (Measurement, Meteorological and Satellite Data) vs. Case B (Measurement and Meteorological Data), we observe that all the models' performance is better in case C, which is evident with lower values for MSE ($<0.009$ $(\mu g/L)^2$), RMSE ($<0.097$ ug/L), MaxError ($<0.103$ µg/L), and MAE ($<0.097$ µg/L) and higher $R^2$ values ($<0.81$); however, LSTM and RNN showed better performance against SARIMAX (Table 5). This suggests that incorporating a larger set of chlorophyll-a-related variables both directly and indirectly enhances the predictive capacity of the algorithms. Good chlorophyll predictions have been obtained in investigations that have used deep learning models and Landsat-8 in lakes, such as the case of [77–79]; as in the case of this research, these techniques can be improved by incorporating meteorological variables. The methodology of this study and other similar methodologies have applications in monitoring water quality and serve as an early warning tool for hydro-environmental management in inland water ecosystems, according to [35,81]. It is important to clarify that the precision in the models will always be greater when more input data are provided. Only available meteorological and satellite data were used in this manuscript. The more images included in the model, the better the estimate should be, and image quality can be affected by cloud density (cloud percentage) as it can alter the pixel value (band or calculated index) and decrease the precision of the model accuracy. The image quality was not a limitation in our work, but the fact of not having some images close to the monitoring due to the high cloud percentage, which prevented the use of some in situ data, was. Generally, in similar investigations, cloudiness can be a limiting factor in the availability and quality of satellite data, and therefore, affect the precision of the estimation of water quality parameters.

The "Ley de Bases del Medio Ambiente" (Ley Nº 19.300 de 1994) in Chile defines aquatic pollution in terms of the existence of standards that establish permissible limits for the presence of substances, elements, or energies, susceptible to causing environmental damage. Lake Llanquihue and the Villarrica side are the only lake systems in Chile that have a secondary water quality standard that seeks to safeguard the use of water resources, protect, and conserve the aquatic communities and ecosystems of the lake, and maximize the benefits that the ecosystem services associated with the lake provide [35]. Therefore, it is of vital importance to maintain a follow-up of these inland aquatic bodies, as Chl-a is a bioindicator parameter of algae presence commonly used in research. It is relevant to inform the authorities and the population of the current state and evolution of the lake through research such as this one. It also provides valuable base information for the management of water resources that provide us with multiple uses. In the future, we intend to use the models tested in the estimation of parameters at times of the year when the conditions of in-situ monitoring of Llanquihue Lake represent a limitation (intense wind or rainy periods). However, these estimation models are relevant in autumn and winter, when multispectral satellite images present a high percentage of cloud cover.

## 5. Conclusions

Combined remote sensing and machine learning techniques have proven to be valuable tools for the estimation of environmental proxies, such as chlorophyll-a. Coefficient of determination values ranging from 0.64 to 0.93 were obtained, with the LSTM model showing the best statistics in any of the cases tested. The LSTM model generally performs well across most stations, with lower values for MSE ($<0.260$ $(\mu g/L)^2$), RMSE ($<0.510$ ug/L), MaxError ($<0.730$ µg/L), and MAE ($<0.442$ µg/L). This parameter has been widely used in different aquatic ecosystems as an indicator of algal biomass and water quality. In this study, a series of in-situ data from 1989 to 2021 recorded at eight monitoring stations

spatially distributed in Llanquihue Lake was used to study the behavior of limnological variables at different points in the lake. The three estimation models employed demonstrated strong performance in estimating Chl-a, with the LSTM model yielding the most accurate results. Of the three cases applied in this study, Case C (all variables integrated), meteorological, water quality measurements, and satellite data showed the most accurate results for all stations in the lake. These models will be employed in future research focused on seasonal periods such as autumn and winter, characterized by frequent episodes of rain or "Puelches" (strong winds). Traditional monitoring methods face increased complexity during these periods. Therefore, as an alternative, recovery models like the ones presented above have emerged, taking advantage of deep learning tools to integrate real-time data with satellite observations, allowing early tools to be developed for monitoring lakes in tracking algal bloom phenomena. In addition, by combining these data sets, these models provide a more effective approach to monitoring and analyzing weather conditions during challenging periods.

**Supplementary Materials:** The following supporting information can be downloaded at: https://www.mdpi.com/article/10.3390/rs15174157/s1. Figure S1. Error Case A, LSTM model. Figure S2. Error Case A RNN network. Table S1. Satellite Images characteristics. Table S2. Behavior of limnological parameters in the Lake Llanquihue.

**Author Contributions:** Conceptualization, L.R.-L.; methodology, L.R.-L. and D.B.U.; software, D.B.U. and I.D.-L.; validation, L.B.A., L.R.-L. and D.B.U.; formal analysis, L.R.-L.; investigation, L.R.-L.; resources, R.U.; data curation, L.R.-L. and D-B-U.; writing—original draft preparation, L.R.-L. and D.B.U.; writing—review and editing, L.R.-L., D.B.U., S.Y., L.B., F.F., I.D.-L. and S.Y.; visualization, L.B.A. and I.D.-L.; supervision, R.U.; project administration, R.U.; funding acquisition, L.R.-L., L.B., F.F. and S.Y. All authors have read and agreed to the published version of the manuscript.

**Funding:** This research was funded by CRHIAM (ANID/FONDAP/15130015) and with the collaboration of the Chilean government through ANID's Fondecyt Regular Project 1221091.

**Data Availability Statement:** The data presented in this study are available upon request from the corresponding author.

**Acknowledgments:** L.R.-L. is grateful to the Centro de Recursos Hídricos para la Agricultura y la Minería (CRHIAM) (Project ANID/FONDAP/15130015) and S.Y. is grateful for ANID's support through the Fondecyt Regular Project 1221091.

**Conflicts of Interest:** The authors declare no conflict of interest.

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
