# Peer review of "Estimation of Water Quality Parameters through a Combination of Deep Learning and Remote Sensing Techniques in a Lake in Southern Chile"

_remotesensing, doi:10.3390/rs15174157_

Round 1

Reviewer 1 Report

This manuscript presents a study to estimate water quality by combining deep learning technique and satellite remote sensing data for lakes in Chile. As expected, three deep learning models performed nearly equally but all perform better when adding remote sensing data in addition to in situ data. The manuscript was written well and can be accepted after some minor corrections.

1.    In line 165, it should be 1.395 um rather than 1395 um.

2.    There is a typo in line 590 ‘tomonitor’.

3.    The sentence in the Supplementary Materials is not finished ’can be downloaded at : ’ 

4.    The discussion section looks like conclusion and it can be merged to conclusion part.

Author Response

Responses to Reviewers

We thank the editor and reviewers for their extended time and the valuable comments, which improve the quality of our manuscript. Our responses are formatted in bold and cursive with the mark “//”. In the manuscript the changes and additions are highlighted in yellow, blue and green colors, following the patterns Reviewer #1: yellow color; Reviewer #2: blue color and Reviewer #3: green color, respectively. Best regards,

Dr. Lien Rodríguez López and authors

Reviewer 2 Report

In this paper, the authors compared the inversion effects of three models(SARIMAX, LSTM, and RNN) on chlorophyll concentration under three input data cases(case A, B, C). I think that this study provides a potentially feasible approach for the inversion of chlorophyll-a concentration and the early warning of algal blooms in inland freshwater lakes around the world.

Main concerns:

1. The combination of deep learning and remote sensing technology is the feature of the algorithm used in this paper. Therefore, it is necessary to introduce the deep learning algorithms (LSTM, RNN) used in more detail, such as the input and output of the network, the number of hidden layers, the number of neurons in each layer, the overall topology diagram, and the specific Activation function. This information is very helpful for readers to understand the specific algorithm of the paper, and it is recommended that the authors supplement it appropriately

2. Training and validation data are very important for deep learning. It is recommended that the authors provide the number of samples for training and validation data, as well as the method of partitioning. Should they be randomly selected or selected based on the data acquisition time?

3. According to the research results of this paper, when meteorological parameters and remote sensing image data are added, the accuracy of chlorophyll concentration estimation in several models has been improved. However, in practical applications, more input data will be needed. In some cases, meteorological parameters may be lacking, and in others, high-quality remote sensing image data may be lacking (such as high cloud coverage). It is recommended that the authors discuss the limitations and practicality of the algorithm used in the paper.

Minor comments:

1. P.2, Line 90: Please provide the full name of the abbreviation “m.a.s.l”, when it first appears in the text.

2. P.7, Line 317: What is the meaning of “softmax” in equation (11)? Please provide an explanation.

3. P.15, Line 512: "Results shown that… ". In this sentence, “shown” should be changed to “showed”.

4. P.15, Line 515: “shown” should be also changed to “showed”.

5. P.15, Line 516: Does “Ph” here refer to the pH value? Please clarify it.

6. P.16, Line 565: “above >0.110” should be replaced with “above 0.110”.

7. P.17, Line 590: “tomonitor” should be “to monitor”. 

Author Response

(The authors gave the same response as above.)

Reviewer 3 Report

Dear authors,

There is disconnection between the device 2.3, 2.4 and 2.5. Section 2.4, which is the computer description, is not adapted to biophysical inputs: reflectance bands, NDVI and FAI. The SARIMAX mathematical model must be adapted, as well as equation 1 to the input variables and using these inputs is the explanation of the process. Once these models with limput have been explained, an output should appear and know how it has been achieved to compare with the field data. Figure 2 should be explained with the input data, acolite bands, NDVI and FAI. All equations must be adapted to the input variables or obtained over time. We do not know the number of images used. In conclusion, the language of the computer tool must adapt to the problem of nature. In line 371 it says that figures 4 and 5 show limnological behavior, when the bottom of figure 4 RNN network architecture with a loop. It has nothing to do with limnological behavior. Figure 5 is not understood, it should be improved in its explanation. Figure 6 shows a temporary series that we do not know where it comes from since the images used or the process used for its generation, etc., have not been indicated at any time. Line 590 is not understood. Discussions should be enhanced by explaining a law of the land. In the conclusions it is not clear that the objective of the paper has been fulfilled; The objective of this study is to contribute through combined techniques of remote sensing and machine learning to develop early tools for monitoring lakes in the follow-up of algal bloom phenomena

Best regards

Author Response

(The authors gave the same response as above.)
